# Influential Factors of Local Tissue Necrosis after Taiwan Cobra Bites: A Secondary Analysis of the Clinical Significance of Venom Detection in Patients of Cobra Snakebites

**DOI:** 10.3390/toxins13050338

**Published:** 2021-05-07

**Authors:** Chih-Chuan Lin, Chung-Hsien Chaou, Shi-Ying Gao

**Affiliations:** 1Department of Emergency Medicine, Lin-Kou Medical Center, Chang Gung Memorial Hospital, Taoyuan 33305, Taiwan; Shien@url.com.tw (C.-H.C.); s78092359@cgmh.org.tw (S.-Y.G.); 2College of Medicine, Chang Gung University, Taoyuan 33302, Taiwan; 3Chang Gung Medical Education Center, Chang Gung Memorial Hospital, Taoyuan 33302, Taiwan

**Keywords:** cobra, snakebites, tissue necrosis, venom, antivenom

## Abstract

Local tissue swelling, inflammation, and wound necrosis are observed in Taiwan cobra bites. Knowledge of the factors influencing local tissue necrosis after cobra bites might improve the cobra bite treatment strategy. Therefore, we aimed to explore the factors influencing local tissue necrosis after cobra bites. This was a retrospective observational cohort study. All patients clinical presentations including serum venom levels for determining the influential factors in this study were obtained from Hung et al.’s previous study. Clinical features, such as bite information, initial swelling, patient presentation time, serum venom levels, and antivenom, use were extracted. The measurement outcome was the development of wound necrosis. The factors influencing wound necrosis were investigated using univariate and logistic regression analyses. The influential factors of local tissue necrosis and their areas under the curve were: initial limb swelling, 0.88; presentation time × serum level, 0.80; initial necrosis, 0.75; patient presentation time, 0.70. Serum venom level alone cannot be used as a predictive factor. The development of tissue necrosis might be associated with the venom factor, time factor, and their interaction. These influential factors can be used in future studies to evaluate antivenom efficacy.

## 1. Introduction

Cobra bites are a critical issue given that snakebite envenomation is considered a neglected tropical disease [1]. Except for systemic neurotoxicity, different degrees of muscle weakness, paralysis, significant local tissue swelling, inflammation, and wound necrosis follow most cobra snakebites [2,3,4,5,6,7]. As one of the six medically important venomous snake species found in Taiwan, Taiwan cobra (*Naja atra*) envenomation accounts for approximately 20% of snakebite cases but with almost no mortalities [8]. When comparing its clinical effects with crotaline snake envenomation (*Viridovipera stejnegeri* and *Protobothrops mucrosquamatus* in Taiwan), Taiwan cobra envenomation causes more serious wound complications, such as local tissue swelling and necrosis (Figure 1), with few neurotoxic symptoms [9,10,11]. Cytotoxins of the three-finger toxin family are thought to cause wound necrosis after cobra bites [12]. It is also accepted that higher cytotoxin doses may induce more extensive tissue necrosis [13].

Physicians use the bivalent freeze-dried neurotoxic antivenom (FNAV) produced by the Centers for Disease Control of Taiwan to treat cobra bites. The proposed roles of this FNAV antivenom treatment are to reduce limb swelling and prevents tissue necrosis. However, even with large antivenom doses, as recommended by the Taiwan poison control center, the prevalence of wound necrosis and surgical procedures, such as debridement, remains high [11,12,13,14,15].

In Hung’s previous study, the authors observed that serum venom concentrations were suitable for differentiating between severe and mild envenomation [16]. However, the authors did not determine the relationship between serum venom concentration and the occurrence of tissue necrosis in their previous work. We hypothesize a correlation between the serum venom level and tissue necrosis development after cobra bites. Based on the aforementioned hypothesis, we investigated the factors influencing (including serum venom level) local tissue necrosis after cobra bites.

## 2. Results

### 2.1. Patient Characteristics

There were 27 patients (6 men and 21 women) enrolled in this study (Table 1). We divided these 27 patients into wound necrosis and no wound necrosis groups according to whether wound necrosis developed or not. The median ages of the wound necrosis and no wound necrosis groups were not significantly different (45.5; IQR 33.5–61.5 vs. 48; IQR 34–59, *p* = 0.99). Except for one patient who was bitten on the face, all patients were bitten on the limbs (17 upper limbs, 9 lower limbs). Systemic symptoms were observed in 16 of the 27 patients (59.2%). Of those 16, 7 patients demonstrated neurological muscle weakness (serum venom level: 228–1270 ng/mL), and 9 patients showed gastrointestinal symptoms. In contrast, 11 patients presented with local symptoms. There was no statistically significant difference between the wound necrosis and no wound necrosis groups in the systemic symptoms (*p* = 0.14 and 0.37 for neurological and gastrointestinal symptoms, respectively).

In total, 18 patients had a severity of initial local tissue swelling of none to a mild degree, and the other nine patients presented with a moderate to severe degree of swelling. In total, eight patients developed local tissue necrosis among the 27 patients. Of the 8 patients, 7 had moderate to severe initial local swelling, whereas 2 of the 19 patients with none to mild initial local swelling developed local tissue necrosis. Initial necrosis was found in four patients when they presented to the emergency department. Their serum venom concentrations and sampling times were as follows: 633 ng/mL/6 h, 228 ng/mL/4 h, 590 ng/mL/13 h, and 986 ng/mL/16 h (Figure 1). These four patients also showed tissue necrosis. Nine patients underwent surgical procedures.

Doctors who treat snakebite patients administer antivenom doses based on their evaluation or follow the 1–6 vials of FNAV recommended by the Taiwan Poison Control Center. Most of the patients (19/27) had blood samples obtained before antivenom administration. The median serum venom level was 228 ng/mL (IQR, 2–810). The average sampling time was 4 h (IQR, 2–16). The median dose of FNAV antivenom was 2 vials (IQR, 1–6), ranging from 0 to 15 vials. Nine patients (cases 1–10, except case 6, Appendix A) with mild envenomed severity had serum venom levels of 0–24 ng/mL before antivenom use, which was significantly lower than that found in patients with severe envenomation (cases 11–19, Appendix A; *p* = 0.0066, Wilcoxon rank-sum test). There were no ICU admissions or any other life-threatening conditions related to the snake bite in these patients.

### 2.2. Influential Factors of Local Tissue Necrosis

#### 2.2.1. Patient Groups According to Presentation Time and Clinical Severity

We divided the patients into three groups according to their presentation times and clinical severity. Group I, the early and mild presentation cases (cases 1–10, except case 7, with a presentation time of 7 h) received antivenom within 6 h (medium, 2.25 h; IQR, 2–4; range, 1 to 7 h; Figure 1). In summary, patients with low venom concentration/load who received antivenom did not develop necrosis.

Group II (cases 11–19) was the patient group with early presentation (≤6 h) with higher serum venom levels and moderate to severe presentation. They received 0–15 vials of antivenom administered between 4 and 6 h. Among them, three cases had severe envenomation and necrosis. One of them did not receive antivenom treatment for unknown reasons, and two received one and five vials of antivenom. In Group II, we observed that five patients with the highest serum venom levels who received 2–8 vials did not develop necrosis (dot circle, Figure 1). However, three of the four cases with lower serum venom levels developed necrosis, and all three cases presented with initial necrosis. Based on these observations, there might be significant interactions between venom load (venom concentration), patient presentation, and antivenom use.

Group III (cases 20–27) comprised late-presentation patients who presented to the emergency department (ED) after more than 12 h. In total, five patients developed tissue necrosis. They received 1–7 vials of antivenom.

#### 2.2.2. Comparison between the Wound Necrosis and No Wound Necrosis Groups and Derived Influential Factors of Cobra Bites Associated with Wound Necrosis

When comparing the wound necrosis and no wound necrosis groups, patients with wound necrosis had more severe initial local swelling (7/8 vs. 2/19; *p* = 0.006), more initial necrosis (4/8 vs. 0/19; *p* = 0.004), and late presentation times (14.5; IQR 5–125.5 vs. 3; IQR 2–6, *p* = 0.02, Table 1). No statistical difference was observed in serum venom levels (372; IQR 153.5–611.5 vs. 21; IQR 0–874, *p* = 0.31) between these two groups. However, the Group II patients, the early and higher serum venom level group, had much higher venom levels, but no tissue necrosis developed (Figure 2). Based on the above analytical results, we assumed that the development of tissue necrosis might be associated with the venom factor, time factor, and their interaction. Thus, we calculated the product of these two factors (presentation time × serum venom level) and used the outcome to clarify their interaction and association in developing tissue necrosis. This relationship is presented in the following equation:serum venom concentration = 2070.4(presentation time) ^−0.8^,or(presentation time)^0.8^ × serum venom concentration = 2070.4(1)

The wound necrosis group had a higher product of presentation time × serum venom level (5.734; IQR 937–15,342.5 vs. 42; IQR 0–2430, *p* = 0.01). The power of presentation time and serum venom level was 0.8 and 1, respectively. The serum venom level had a larger impact on tissue necrosis development than the patient’s presentation time.

A generalized form of Equation (1) is as follows:f (tissue necrosis) = presentation time^n^ × serum venom concentration = κ(2)
where κ= constant, *n* > or < 1 needs to be determined for each *Naja* spp. bites.

#### 2.2.3. Development of the Clinical Prediction Rule for Wound Necrosis

Clinical characteristics with a *p*-value < 0.1 were further analyzed through multiple logistic regression to identify independent predictors of wound necrosis. Initial local swelling was the only variable that remained in the model after logistic regression. We compared the area under the receiver operating characteristic (ROC) curves of these predictors (Figure 3). The areas under the curve of these predictors were initial local swelling, 0.88; time × serum level, 0.80; initial necrosis. 0.75, and presentation time ≤ 6 h, 0.70. Table 2 displays the sensitivities, specificities, positive predictive values, negative predictive values, and accuracies of the above variables in predicting the development of tissue necrosis.

## 3. Discussion

### 3.1. Influential Factors in Predicting the Development of Local Tissue Necrosis

In this study, we identified initial local limb swelling, time × serum level, initial necrosis, and the patient presentation time as the influential factors for predicting the development of local tissue necrosis after cobra bites. Following antivenom administration, the necrotic tissue in patients who initially presented with necrosis was not resolved. The most potent and straightforward predictor was moderate to severe initial local limb swelling. These initial factors can be used to assess whether antivenom can prevent tissue necrosis development.

We found that serum venom concentration, patient presentation time, and the interaction between them were influential factors in tissue necrosis development after cobra bites. The simplest explanation of the presentation time × serum venom level effect is that the longer the presentation time, the higher the serum venom level, the greater the possibility of necrosis formation after cobra bites. For example, the Group I patients presented early (≤6 h) and had low serum venom concentration; there was no tissue necrosis development after antivenom administration. Furthermore, cases 23 and 24 had the same venom level (2 ng/mL) at presentation to the ED; however, case 23 developed necrosis, but case 24 did not because case 23 had a higher venom load and longer venom effect time than case 24.

Nevertheless, case 11 merits our attention. Case 11 presented to the ED as early as 2 h after the cobra bite but received no antivenom, and necrosis developed at a venom level of 481 ng/mL. Thus, when compared with other cases with higher venom concentrations with no tissue necrosis in Group II, the use of antivenom, or the lack thereof, was considered as one of the determining factors in tissue necrosis development.

### 3.2. Patient Presentation Time as an Influential Factor of Cobra-Bite Associated Wound Necrosis

A review of 292 *Naja atra* envenomated patients revealed that patients admitted within 12 h of the snakebites had better outcomes than those admitted 12 h or more after the snakebites [17]. Sarava et al. suggested that delay in presentation to the hospital was a risk factor associated with poor outcomes in *Naja naja* snakebite patients [18]. In Tilbury’s study, antivenom efficacy was indirectly proportional to the time elapsed between the bite and antivenom administration [5].

In the present study, all patients except three patients who received antivenom within 6 h did not develop tissue necrosis. Two cases of tissue necrosis were well established despite their early presentation to the ED, and they eventually developed a larger necrotic area. Case 11 presented to the ED as early as 2 h after the cobra bite but did not receive antivenom, and tissue necrosis developed. The above observation corroborates previous animal trials [12,19], where it was demonstrated that necrosis development occurred as early as 3 or 6 h after envenomation with *Naja nigricollis* venom. Therefore, we introduced the concept of necrosis time. We defined necrosis time as the time tissue necrosis develops without antivenom administration at a specific venom concentration. Thus, the necrosis time for case 11 was 2 h at a venom level of at least 481 ng/mL. Case 11 and early presentation cases with initial necrosis indicate that tissue necrosis might be inevitable in some cobra bite cases at presentation.

As we observed in this study, the venom serum level is typically low in late presentation cases, and necrosis is already evolving. Therefore, determining the time point at which wound necrosis begins to occur is crucial. Here, we suggest a venom administration time of at most 6 h after snakebites.

### 3.3. Venom Level (Venom Load) As an Influential Factor of Cobra Bite Associated Wound Necrosis

Higher venom levels imply a higher venom load delivered to snakebite victims. For example, the patients in Group I did not develop necrosis because they were mildly envenomated. A recent animal study indicated that more necrotic tissue developed with more cytotoxins administered to mice [13]. However, we should consider the patient’s presentation time when interpreting the serum venom level to predict tissue necrosis development. We can use Equation (2) in such a situation. Using Equation (2), we can anticipate the inevitability of tissue necrosis. These data can be used to assess antivenom efficacy in preventing tissue necrosis development in the future.

### 3.4. Possible Clinical Applications of Our Study Findings in other Naja *spp.* Bites

A cobra bite is a critical issue as snakebites envenomation is considered a neglected tropical disease [1]. Envenomation caused by different *Naja* species such as *Naja siamensis* (Thai spitting cobra) [4], *Naja kaouthia* (monocellate cobra) [2,20], *Naja naja* (Indian cobra) [3,21], and *Naja mossambica* (Mozambique spitting cobra) [5,7] share the common features of local tissue swelling, inflammation, infection, and significant tissue necrosis as does Taiwan cobra bites [11,14]. Our study has substantial potential for improving treatment strategies for cobra-bite-associated wound necrosis. Our study offers the medical community a framework for stratifying the risk of developing wound necrosis following a snakebite, which can be improved upon and refined further for each *Naja*. spp.

### 3.5. Limitations

Our study has some limitations. The first limitation of this study is that the equations proposed here are yet to be validated. Second, our sample size was small. We calculated the statistical power to ensure adequate power to address this hypothesis. The sample size needed to compare the wound necrosis and no wound necrosis groups using an independent *t*-test was calculated with a presumed effect size of 0.6 under a two-tailed comparison, a preset α-level of 0.05, and a power of 0.8. Third, antivenom and dose administration were important interference factors in determining the occurrence of tissue necrosis in this study. We had no way to eliminate any interference in this study. A well-designed animal trial is needed to clarify the role of the patient presentation time and serum venom level in the development of tissue necrosis in cobra bites. Fourth, this is a secondary analysis of retrospectively collected data; therefore, the study design has certain inherent limitations; thus, we should interpret the results cautiously. Any “cause and effect” conclusions cannot be made. Fifth, the data we used here are almost 2 decades old, and this will raise the question of whether we can use these data presently. However, Hung and colleagues used a similar ELISA method recently [22].

In the present study, it was not established whether patients received local antiseptics as part of their treatment. At present, whether local antiseptics play a role in the development of wound necrosis is unknown. However, if wound infection occurs, antibiotics are administered. Thus, wound infection would not be a significant contributing factor to cobra-bite-related wound necrosis. Antivenom treatment (dose and time of administration) may influence local necrosis development. However, antivenom is necessary for managing patients with venomous snakebites; since antivenom administration is inevitable, it may be one of the limitations of this study.

Finally, there was only one male patient in the wound necrosis group and five male patients in the no wound necrosis group. This was likely because of the small sample size used in the present study. Whether gender has an impact on necrosis development requires further research.

## 4. Conclusions

In this study, we identified four factors that influence local tissue necrosis after Taiwan Cobra bites. We also proposed the interaction of venom and time (expressed as presentation time × serum level) in wound necrosis development. The influential factors were initial limb swelling, initial necrosis, patient presentation time, and the presentation time × serum level interaction. Among these, initial moderate to severe limb swelling was the most potent variable for predicting tissue necrosis development after cobra bites. Animal studies have shown that venom (or cytotoxins) concentration might be positively correlated with the likelihood of necrosis development. However, serum venom level alone cannot be used as a predictive factor. We should consider the patient’s presentation time as another factor in wound necrosis development. These influential factors could be used in future studies to evaluate antivenom efficacy. Based on this study, we also suggest a venom administration time of at most 6 h after snakebites.

## 5. Materials and Methods

### 5.1. Settings, Patients, and Data Collection

This was a secondary analysis of a retrospective cohort study. According to Hung et al.’s study, all patients enrolled in this study were admitted to the ED of Taichung Veterans General Hospital, Taiwan, between 1994 and 1998. There were 27 patients (6 men and 21 women) enrolled in the study. Their median age was 48 years (IQR 34–59).

We divided these 27 patients into wound necrosis and no wound necrosis groups depending on whether wound necrosis developed. We retrieved all the patients and their clinical presentations from Table 1, Table 2 and Table 3 of Hung’s published article: demographic (age and sex), snakebite information (the culprit snake, time of biting), clinical symptoms (the location and degree of limb swelling and their progression), sampling time/serum venom levels, and antivenom treatment (time of antivenin administration and dose). Appendix A contains all the retrieved pieces of information that we used to analyze the influential factors of local tissue necrosis.

The institutional review board (IRB) of Chang Gung Memorial Hospital (approval no: 202001415B0, date of approval: 2020/08/24) approved this study.

### 5.2. The Severity of Local Manifestations of Envenomation

Since there is no consensus on the severity grading for snakebite envenomation in Taiwan at present, we defined the severity of local manifestations of envenoming as mild, moderate, and severe according to a previous publication (Table 3). Systemic effects include nausea, vomiting, abdominal pain, fasciculations, muscle weakness, tachycardia, hypotension, incontinence, epistaxis, hematuria, coagulopathy, hemolysis, renal failure, and cardiopulmonary arrest [16,23]. According to Hung et al. [16], we defined early presentation patients as those who arrived at the ED within 6 h after the snakebite. We defined late-presentation patients as those who arrived at the ER more than 6 h after the snakebite.

### 5.3. Study Protocol and Data Analysis

Patients were divided into wound necrosis and no wound necrosis groups. To investigate the influential factors of local necrosis and test our hypothesis, univariate and logistic regression analyses were performed. We also produced a figure of patient distribution with details of venom levels, presentation time and wound condition (necrosis or no necrosis) to better understand the role of the time factor in the development of tissue necrosis (Figure 2).

### 5.4. ELISA for Taiwan Cobra Venom

Here is a summary of the ELISA method developed by Hung et al. [16]. *N. atra* venom and specific antivenom were obtained from the Vaccine Centre, Taiwan CDC. The venom was lyophilized and stored at 4 °C.

Antivenom was coated on the polystyrene surface of 96-wells plates with 100 mL of 100 mM sodium carbonate buffer (pH 9.6) containing 20 mg/mL rabbit antivenom at 4 °C overnight. The washing procedure was as follows: the plates were first washed five times with PBS (containing 0.05% Tween-20, pH 7.6). They were then blocked using a 1% gelatin PBS solution. Afterward, 0.1 mL of the calf serum diluted test samples were added to each well and incubated for 1 h at 37 °C, followed by further washing. Subsequently, 100 mL of 100 mM sodium carbonate buffer (pH 9.6) containing 20 mg/mL of equine antivenom was added to each well and incubated for 2 h at 37 °C.

After washing, 25 µg horseradish peroxidase-coated goat anti-rabbit immunoglobulin antibody (in 100 mM sodium carbonate buffer, pH 9.6) was added to each well and incubated for 1 h at 37 °C. After washing, a 2 mg/mL substrate solution of O-phenylenediamine in 10 mM sodium phosphate buffer (pH 7.3, containing 0.06% H_2_O_2_) was added to the wells. The color reaction was stopped using 100 mL of 2N H_2_SO_4_ solution.

The plate was read using a spectrophotometer (Spectra and Rainbow Reader; SLT Lab Instruments, Salzburg, Austria) at 490 nm absorbance. The results were obtained by subtracting the absorbance readings of the blank well. The standard assay was performed on the same plate as the venom detection assay.

The ELISA method performed well since the standard curve had an excellent linear relationship (*r* = 0.9779) ranging from 1 to 100 ng/mL between cobra toxin concentration and optic density at 492 nm.

### 5.5. Statistics

Categorical variables were reported as frequencies and percentages, while continuous variables were expressed as medians (±IQR). Univariate analyses were conducted using the Mann-Whitney U test for numerical variables and the chi-squared test for categorical variables, with odds ratios calculated to assess the strength of association. We chose variables with *p* < 0.1 for further multivariate analysis using multiple logistic regression with stepwise regression and no interaction term. The serum venom level × presentation time analysis was determined prior. The receiver operating characteristic (ROC) curve was plotted. An area under the curve (AUC) analysis was performed to examine the accuracy of variables, with cut-off points identified with sensitivity and specificity using the Youden test. Statistical significance was set at *p* < 0.05. All statistical analyses were performed using SAS statistical software version 9.4 (SAS Institute, Cary, NC, USA).

## Figures and Tables

**Figure 1 toxins-13-00338-f001:**
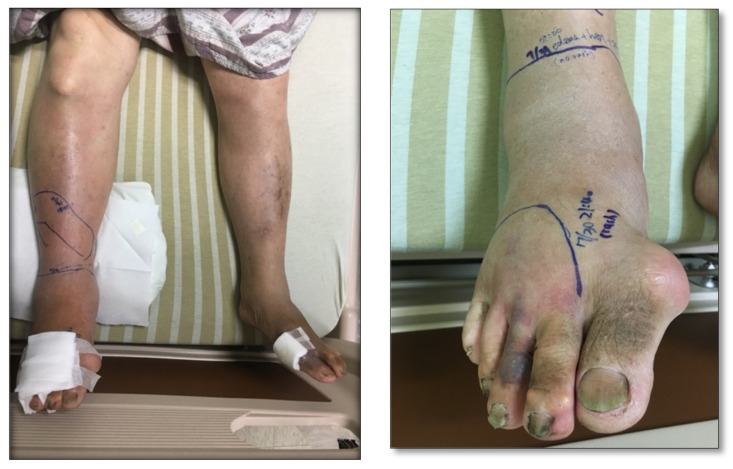
An example of limb swelling and wound necrosis after Taiwan cobra bites. (**Left**): lower leg swelling progressively to the right knee joint. (**Right**): a necrosis wound was observed at the 3rd toe.

**Figure 2 toxins-13-00338-f002:**
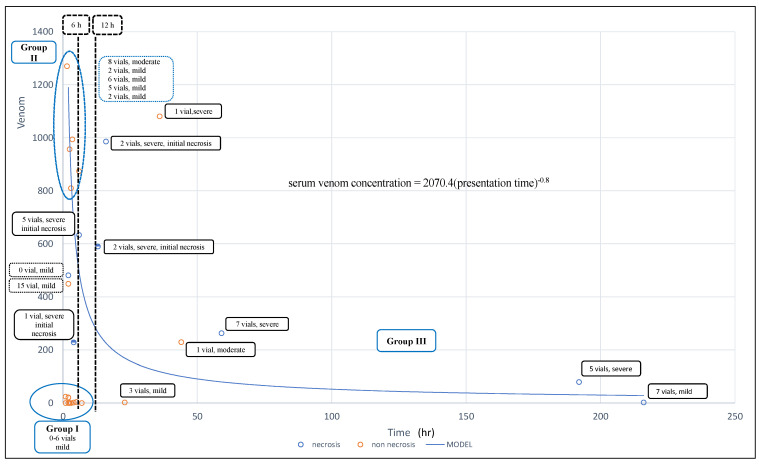
The fitted curve for patient presentation time against serum venom among patients who developed wound necrosis and their clinical information. (1) Cases on the left bottom belong to case Group I (early and mild presentation). No tissue necrosis developed in this group. (2) Cases on the left side of the 6-h line are Group II patients. Patients who presented early and with worse local manifestations. Six of them did not develop wound necrosis. Five cases on the upper left (within the circle) had high serum venom levels but did not develop necrosis after antivenom administration. However, three of them, including one case that did not receive antivenom, developed wound necrosis. The necrotic lesions may occur as early as 4–6 h after the snakebite. (3) Most of the Group III cases who presented late developed tissue necrosis despite antivenom treatment. They had low serum venom levels.

**Figure 3 toxins-13-00338-f003:**
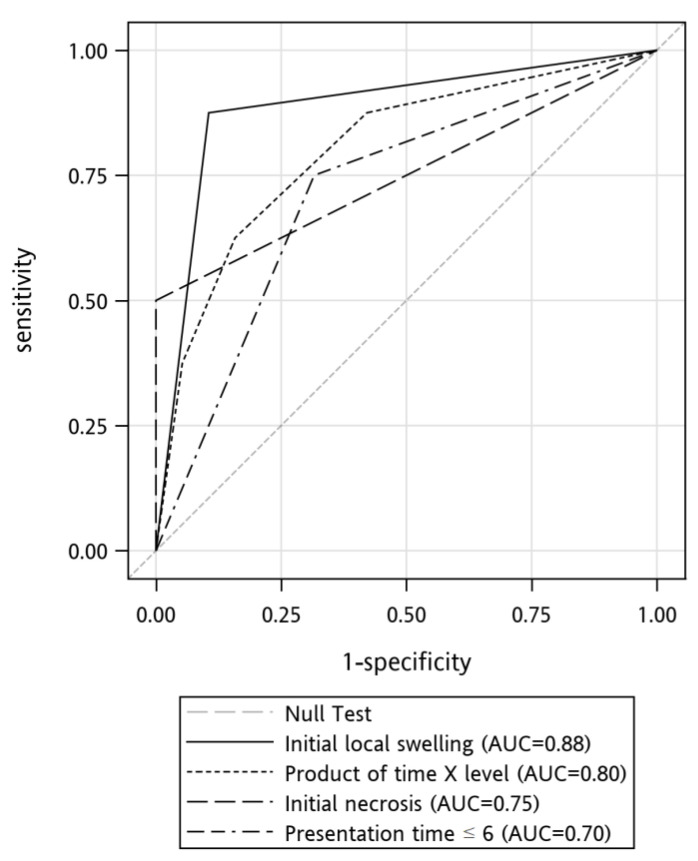
Receiver operating characteristic (ROC) curves for the prediction of necrosis development.

**Table 1 toxins-13-00338-t001:** Clinical characteristics and their comparisons between wound necrosis and no wound necrosis groups.

Variables	Wound Necrosis Group *n* = 8; *n* (%)	No Wound Necrosis Group *n* = 19; *n* (%)	*p*
Patient characteristics			
Age ^#^	45.5 (33.5–61.5)	48 (34–59)	0.99
Male sex	1 (12.5)	5 (26.3)	0.90
Bite information			
Lower extremity	5 (62.5)	12 (63.2)	0.79
Upper extremity	3 (37.5)	6 (31.6)	
Face	0 (0.00)	1 (5.3)	
Fang mark			0.09
0	0 (0.00)	3(15.8)	
1	0 (0.00)	5 (26.3)	
2	8 (100)	11 (57.9)	
Clinical Features			
Initial local swelling			0.006
None to mild	1 (12.5)	17 (89.5)	
Moderate to severe	7 (87.5)	2 (10.5)	
Initial necrosis	4 (50)	0 (0.00)	0.004
Systemic Symptoms/signs			
No	0 (0.00)	11 (57.9)	
Neurologic (muscle weakness)	3 (37.5)	4 (21.1)	0.14
Gastrointestinal	5 (62.5)	4 (21.1)	0.37
Antivenom before sampling			0.01
0 vial	3 (37.5)	16 (84.2)	
1 vial	5 (62.5)	2 (10.5)	
2 vials	0 (0.00)	1 (5.2)	
Serum venom level (ng/mL) ^#^	372 (153.5–611.5)	21 (0–874)	0.31
Presentation time (h) ^#^	14.5 (5–125.5)	3 (2–6)	0.02
Product of time × level ^#^	5734 (937–15,342.5)	42 (0–2430)	0.01
Treatment Modalities			
Total antivenom (vial)	3.5 (1.5–6)	2 (1–6)	0.52
Operation needed	8 (29.6)	1 (3.7)	<0.0001

^#^ these variables were expressed as Median(Q1–Q3). Other variables were expressed as count and the percentage.

**Table 2 toxins-13-00338-t002:** Sensitivities, specificities, positive predictive values, negative predictive values, and accuracies in predicting the development of tissue necrosis.

Variable	Sensitivity (%)	Specificity (%)	PPV ^#^ (%)	NPV ^#^ (%)	Accuracy (%)
Initial local swelling	87.50	89.47	77.78	94.44	88.89
time × serum level (Cut-off point 3500)	62.50	84.21	62.50	84.21	77.78
Initial necrosis	50.00	100	100	82.61	85.19
Presentation time	62.50	78.95	55.56	83.33	74.07

^#^ PPV/NPV: Positive and Negative Predictive Values.

**Table 3 toxins-13-00338-t003:** Definitions of the severity of envenoming.

Envenoming Severity	Definition
dry bite	no swelling or erythema around the fang marks
mild degree	limb swelling, erythema limited to one joint area, or was equal to or less than 10 cm in size
moderate degree	swelling and erythema around the fang marks were 10–20 cm and/or small necrotic change (less than 2 cm in diameter)
Severe degree	limb swelling greater than 20 cm, extended over two joints, and/or significant local tissue necrosis

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
