# Peer review of "Influential Factors of Local Tissue Necrosis after Taiwan Cobra Bites: A Secondary Analysis of the Clinical Significance of Venom Detection in Patients of Cobra Snakebites"

_toxins, 2021, doi:10.3390/toxins13050338_

Round 1
Reviewer 1 Report
Influential factors of local tissue necrosis after Taiwan Cobra bites: A Secondary Analysis of the Clinical Significance of Venom Detection in Patients of Cobra Snakebite In the present review, the authors investigated the influential factors (including serum venom level) of local tissue necrosis after Taiwan cobra (Naja atra) bites. In my opinion, the study is interesting and innovative, including was well delineated. However, I have some comments: Comment (1): General proofreading is highly recommended. Comment (2): Abstract is a little bit confusing and I recommend to the authors to re-write it again. - Also, the abstract missing the aim of the work although the work is very interested and important for readers and clinicians. Comment (3): Introduction. It is very short and missing a lot of information about the snake, available antivenom, and which toxins produce necrosis. Comment (4): Results. It needs a proofreading. - Lines 45-46: could you please clarify what are you meaning by necrosis and no necrosis group? - The same in lines 51-52. - Figure 1. The quality is very low. - I recommend the authors to re-present Table-1 again because it is confusing in the current version. - I am wondering that there are not any photos for the patient show the necrosis or the swelling.Author Response
Dear editor and reviewers,
Thank you for giving us the chance to revise our work "Influential factors of local tissue necrosis after Taiwan Cobra bites: A Secondary Analysis of the Clinical Significance of Venom Detection in Patients of Cobra Snakebites" (manuscript number: toxins-1194720). In this revised edition, we made some changes.
- We re-write the abstract, table 1, the ELISA method, and the conclusion.
- We also put more information of Taiwan cobra and its related venom-induced tissue necrosis in the introduction.
- We added a patient's photos of Taiwan cobra bites limb swelling and wound necrosis as Fig.1, too.
- Reviewer's comments were answered point by point.
- We asked to Editage have extensive English editing of this manuscript.
point-to-point responses to Reviewer 1
Comment (1): General proofreading is highly recommended.
Response: the revised version was proofreading by editage (by CACTUS).
Comment (2): Abstract is a little bit confusing and I recommend to the authors to re-write it again. - Also, the abstract missing the aim of the work although the work is very interested and important for readers and clinicians.
Response: We re-write the abstract as below:
" Local tissue swelling, inflammation, and wound necrosis are observed in Taiwan cobra bites. Knowledge of the factors influencing local tissue necrosis after cobra bites might improve the cobra bite treatment strategy. Therefore, we aimed to explore the factors influencing local tissue necrosis after cobra bites. This was a retrospective observational cohort study. All patients and their clinical presentations/variables and serum venom levels included for determining the influential factors in this study were obtained from Hung et al.’s previous study. Clinical features such as bite information, initial swelling, patient presentation time, serum venom levels, and antivenom use were extracted. The measurement outcome was the development of wound necrosis. The factors influencing wound necrosis were investigated using univariate and logistic regression analyses. The influential factors of local tissue necrosis and their areas under the curve were: initial limb swelling, 0.88; presentation time × serum level, 0.80; initial necrosis, 0.75; patient presentation time, 0.70. Serum venom level alone cannot be used as a predictive factor. The development of tissue necrosis might be associated with the venom factor, time factor, and their interaction. These influential factors can be used in future studies to evaluate antivenom efficacy."
Comment (3): Introduction. It is very short and missing a lot of information about the snake, available antivenom, and which toxins produce necrosis.
Response: You can find the available antivenom in the introduction section. We add some information about the snake and the toxin that cause the tissue necrosis as below.
" As one of the six medically important venomous snake species found in Taiwan, Taiwan cobra (Naja atra) envenomation accounts for approximately 20% of snakebite cases but with almost no mortalities [8]. When comparing its clinical effects with crotaline snake envenomation (Viridovipera stejnegeri and Protobothrops mucrosquamatus in Taiwan), Taiwan cobra envenomation causes more serious wound complications, such as local tissue swelling and necrosis (fig. 1), with few neurotoxic symptoms [9-11]. Cytotoxins of the three-finger toxin family are thought to cause wound necrosis after cobra bites [12]. It is also accepted that higher cytotoxin doses may induce more extensive tissue necrosis [13]."
Comment (4): Results. It needs a proofreading.
- Lines 45-46: could you please clarify what are you meaning by necrosis and no necrosis group?
Response: we add the sentences "We divided these 27 patients into wound necrosis and no wound necrosis group according to whether wound necrosis development or not."
- The same in lines 51-52.
Response: "the wound necrosis and no wound necrosis group" was instead of "the necrosis and no necrosis group'
- Figure 1. The quality is very low.
Response: We did fig. according to the journal's requirement.
- I recommend the authors to re-present Table-1 again because it is confusing in the current version.
Response: Thank you and we re-present table 1.
- I am wondering that there are not any photos for the patient show the necrosis or the swelling.
Response: We add photos for the patient showing the necrosis and the swelling in the introduction section.
Reviewer 2 Report
Authors seek to publish a study titled “Influential factors of local tissue necrosis after Taiwan Cobra bites: A Secondary Analysis of the Clinical Significance of Venom Detection in
Patients of Cobra Snakebite”.
The study did not present any new data, but analyzed some old data to draw some conclusions, that may not be accurate for the present time. The study lacks sufficient control groups to verify the results. The manuscript should be organized and written better, needing improvement in language as well.
Most of my concerns are already included in the section Limitation of the manuscript. Since authors are aware of their limitations, they are encouraged to fix/include all of those missing points before they resubmit this manuscript. Besides, some other specific points are mentioned below.
Specific comments
- What is death rate for Naja atra bite in Taiwan?
- Method is not sufficient & clear, for example ELISA. The method of ELISA needs be improved, particularly 1st paragraph, in terms of buffer, concentrations of venom sample, antivenom, temperature, time of incubation, source of antibodies …etc. Please follow the reference 14 carefully to write this section providing details.
- Authors should mention the number of patients, their age and sex in methods in details.
- It is not clear how neurological muscle paralysis and gastrointestinal symptoms were studied.
- No figure has been provided to show the swelling/wound/necrosis and/or other local/systemic symptoms in patients. Authors need to show some actual images of their observations with explanation.
- It is not clear how the dose of antivenom was determined to be used. Authors should specifically mention the actual concentrations of the antivenom used (instead of number of vials). Authors should perform dose-response of antivenom.
- The number of cases treated/observed in groups (e.g., group ii) is small.
- It is not clear why “One of them received no antivenom”. Is it a control? The number of control is not enough. Please explain.
- It is not clear why and based on what authors defined early presentation as those who arrived at the emergency room (ER) within six hours and late presentations as those who arrived more than 6 hours after snakebite. Six-hour itself is a long period of time for snakebite cases.
- Conclusion is not clear and does not seem to convey any significance of the study.
- Page 232. Provide references to Hung’s published article
- Page 242. The word “therefore” is redundant.
- Page 258, please complete “Hung et” as “Hung et al”
- 259, please correct the word spelling “Plats” to “Plates”
- Check grammar and spelling throughout the manuscript.
Author Response
Dear editor and reviewers,
Thank you for giving us the chance to revise our work "Influential factors of local tissue necrosis after Taiwan Cobra bites: A Secondary Analysis of the Clinical Significance of Venom Detection in Patients of Cobra Snakebites" (manuscript number: toxins-1194720). In this revised edition, we made some changes.
- We re-write the abstract, table 1, the ELISA method, and the conclusion.
- We also put more information of Taiwan cobra and its related venom-induced tissue necrosis in the introduction.
- We added a patient's photos of Taiwan cobra bites limb swelling and wound necrosis as Fig.1, too.
- Reviewer's comments were answered point by point.
- We asked t Editage o have extensive English editing of this manuscript.
point-to-point responses to reviewer 2
- What is death rate for Naja atra bite in Taiwan?
Response: the death rate for Naja atra bite is almost zero in Taiwan. We put the following sentences in introduction: "…. Taiwan cobra (Naja atra) envenomation accounts for about 20 % of snakebite cases but with almost no mortality in Taiwan [8]."
- Method is not sufficient & clear, for example ELISA. The method of ELISA needs be improved, particularly 1st paragraph, in terms of buffer, concentrations of venom sample, antivenom, temperature, time of incubation, source of antibodies …etc. Please follow the reference 14 carefully to write this section providing details.
Response: We re- write the ELISA method as follow.
“N. atra venom and specific antivenom were obtained from the Vaccine Centre, Taiwan CDC. The venom was lyophilized and stored at 4 °C.
Antivenom was coated on the polystyrene surface of 96-wells plates with 100 mL of 100 mM sodium carbonate buffer (pH 9.6) containing 20 mg/mL rabbit antivenom at 4 °C overnight. The washing procedure was as follows: the plates were first washed five times with PBS (containing 0.05% Tween-20, pH 7.6). They were then blocked using a 1% gelatin PBS solution. Afterward, 0.1 mL of the calf serum diluted test samples were added to each well and incubated for 1 h at 37 °C, followed by further washing. Subsequently, 100 mL of 100 mM sodium carbonate buffer (pH 9.6) containing 20 mg/mL of equine antivenom was added to each well and incubated for 2 h at 37 °C.
After washing, 25 µg horseradish peroxidase-coated goat anti-rabbit immunoglobulin antibody (in 100 mM sodium carbonate buffer, pH 9.6) was added to each well and incubated for 1 h at 37 °C. After washing, a 2 mg/mL substrate solution of O-phenylenediamine in 10 mM sodium phosphate buffer (pH 7.3, containing 0.06% H2O2) was added to the wells. The color reaction was stopped using 100 mL of 2N H2SO4 solution.
The plate was read using a spectrophotometer (Spectra & Rainbow Reader; SLT Lab Instruments, Salzburg, Austria) at 490 nm absorbance. The results were obtained by subtracting the absorbance readings of the blank well. The standard assay was performed on the same plate as the venom detection assay.
The ELISA method performed well since the standard curve had an excellent linear relationship (r = 0.9779) ranging from 1 to 100 ng/mL between cobra toxin concentration and optic density at 492 nm.
”
- Authors should mention the number of patients, their age and sex in methods in details.
Response: We had the following sentences in the method section: "There were 27 patients (6 males, 21 females) enrolled in this study. Their median age was 48 (IQR 34-59) years old."
- It is not clear how neurological muscle paralysis and gastrointestinal symptoms were studied.
Response: (1) Different degree of muscle weakness even paralysis can be observed in cobra bites patients. We made this changes in the 2nd lines of introduction. And, made changes from "paralysis" to "muscle weakness "in the 7th lines of section 2. 1.. There was no muscle paralysis in this study. (2) nausea, vomiting, abdominal pain are the common GI symptoms in cobra bites patient
- No figure has been provided to show the swelling/wound/necrosis and/or other local/systemic symptoms in patients. Authors need to show some actual images of their observations with explanation.
Response: We add photos for the patient showing the necrosis and the swelling in the introduction section.
- It is not clear how the dose of antivenom was determined to be used. Authors should specifically mention the actual concentrations of the antivenom used (instead of number of vials). Authors should perform dose-response of antivenom.
Response: (1) Doctors who treat the snakebites patients administrated antivenom doses by their evaluation or followed the suggestion of 1-6 vials of FNAV by Taiwan Poison Control Center. The above sentence was added in the 3rd paragraph in section 2. 1..
(2) Sorry, we cannot provide the information you ask for. There was no such kind of antivenom concentrations reported in Hung et al.'s original paper.
- The number of cases treated/observed in groups (e.g., group ii) is small.
Response: Group II patients was not treated/observed cases. Group II patients was patients who were with early presentation and worse local manifestations. In fig. 2, we addressed "……. (2) Cases on the left side of the 6-hour line are Group II patients. Patients who presented early and with worse local manifestations. ….."
- It is not clear why "One of them received no antivenom". Is it a control? The number of controls is not enough. Please explain.
Response: This was a retrospective cohort study, therefore, there was no control group in this study. One patient did not receive any antivenom (In which reason was not clear) when he was in hospital
- It is not clear why and based on what authors defined early presentation as those who arrived at the emergency room (ER) within six hours and late presentations as those who arrived more than 6 hours after snakebite. Six-hour itself is a long period of time for snakebite cases.
Response: Yes, I agree with you. Some think 6-hour itself is a long period of time for snakebite cases. We defined 6-hour as the early presentation following Hung et al.'s original study design. We add the following sentences in section 5. 2.. " According to Hung et al. [16], we defined early presentation patients as those who arrived at the ED within 6 h after the snakebite. We defined late-presentation patients as those who arrived at the ER more than 6 h after the snakebite."
- Conclusion is not clear and does not seem to convey any significance of the study.
Response: We re-write the conclusion.
" In this study, we identified four factors that influence local tissue necrosis after Taiwan Cobra bites. We also proposed the interaction of venom and time (expressed as presentation time × serum level) in wound necrosis development. The influential factors were initial limb swelling, initial necrosis, patient presentation time, and the presentation time × serum level interaction. Among these, initial moderate to severe limb swelling was the most potent variable for predicting tissue necrosis development after cobra bites. Animal studies have shown that venom (or cytotoxins) concentration might be positively correlated with the likelihood of necrosis development. However, serum venom level alone cannot be used as a predictive factor. We should consider the patient’s presentation time as another factor in wound necrosis development. These influential factors could be used in future studies to evaluate antivenom efficacy. Based on this study, we also suggest a venom administration time of at most 6 h after snakebites."
- Page 232. Provide references to Hung's published article
Response: We already provided it. Hung is the corresponding author of Ref 22.
- Page 242. The word "therefore" is redundant.
Response: "therefore" was deleted.
- Page 258, please complete "Hung et" as "Hung et al"
Response: done.
- 259, please correct the word spelling "Plats" to "Plates"
Response: done.
- Check grammar and spelling throughout the manuscript.
Response: the revised version was proofreading by editage (by CACTUS)
Round 2
Reviewer 1 Report
The Authors addressed all my previous comments in this improved version. I recommend to accept this MS in the present form.
Reviewer 2 Report
The authors have responded to the comments and improved the manuscript.
This manuscript is a resubmission of an earlier submission. The following is a list of the peer review reports and author responses from that submission.
Round 1
Reviewer 1 Report
Please see attached word document.

Reviewer 2 Report
The manuscript clearly needs to be improved both in language and in the way that data is presented. There are also many redundancies in results and discussion which can be avoided.
Reviewer 3 Report
We read with great interest the manuscript untitled “Influential factors of local tissue necrosis after Taiwan Cobra bites: A Secondary Analysis of the Clinical Significance of Venom Detection in Patients of Cobra Snakebite”
In this retrospective study, the authors tried to find prognostic factors of necrosis after Naja atra bites. They studied 27 patients bitten by Naja atra for whom serum venom level was measured. The authors built a formula including time to presentation and serum venom level to predict the development of necrosis. Some issues should be addressed to increase the quality of the manuscript.
A.Results
1-Is there any treatment that might have influenced the development of local necrosis.
Steroids? Anticoagulation? Local anesthetics? local antiseptics?
2-Is there any patients related factors that might have influenced the development of local necrosis: comorbidities like diabetes, immunocompromised status, arteritis…
3-Is there any life-threatening condition related to the snake bite? ICU admission?
4- ROC curve analysis of the presentation time alone should be drawn and compared to the ROC curve of the presentation time X antivenom concentration.
5 -The formula including presentation time and antivenom concentration should include the number of vials to increase its accuracy (if not possible, explain why).
B.Discussion
The discussion is interesting and well written. We suggest adding few points:
We recommend to the authors using the multivariate analysis to try to build a score that might predict necrosis development and to prospectively try to validate the score. It seems more accurate in a clinical setting than a complex formula using time and level of venom. This point should be discussed as it adds perspective to the study. Moreover, is seems that for late presentation, venom serum level was useless as it is usually low, and necrosis already evolving. This point should be discussed.
In the limitation part, the fact that the majority of the patients were female should be discussed (only 1 male in the necrosis group).
C.Table and figures
Figure 1
Interesting figure but too complex. needs simplification.
Figure 2
The legend is not clear. What is the tested factor?
Table 1
Because of the limited sample size results should be presented In median (IQR).
The figures should be revised:
Ligne 6: 5 patients in the non-necrosis group represent 5/8 =62.5 % not 18.5%.
Same: If there are only 8 patients in the necrosis group and if 8 have needed surgery that represents 100% not 29.6%.
It seems that a large part of the patients especially in the no necrosis group have presented systemic signs other than neurological or gastrointestinal. What signs: hemodynamic impairment? Bleeding? Is there any life threatening signs?
- Statistics
The normal distribution of the values should be determined before using a t-test.
However, because of the limited sample size we recommend:
-to present data as median (± IQR) not as mean (±SD)
-to use a non-parametric test (eg. Mann-Whitney U test) instead of student t test
-Youden test should be used to assess best Se and Spe for the ROC curve